# Factors associated with health-seeking behavior amongst children in the context of free market: Household study in Ouagadougou, Burkina Faso, 2011

Idrissa Beogo[1,2,3]*, Drissa Sia[4,5], Patricia Bourrier[6], Darcelle Vigier[6], Nebila Jean-Claude Bationo[7], André Côté[8], Eric Tchouaket Nguemeleu[4,9]

1 École des sciences infirmières, School of Nursing, Faculty of Health Sciences, University of Ottawa, Ottawa, Ontario, Canada, 2 Managua (CIES UNAN-Managua), National University Autonomous of Nicaragua, Managua, Nicaragua, 3 Institut du Savoir Montfort, Ottawa, Ontario, Canada, 4 Département des sciences infirmières, Université du Québec en Outaouais, Saint-Jérôme, Québec, Canada, 5 Département de médecine sociale et préventive, École de santé publique, Université de Montréal, Montréal, Québec, Canada, 6 École des sciences infirmières et des études de la santé / School of Nursing and Health Studies, Université de Saint-Boniface, Winnipeg, Manitoba, Canada, 7 Département d'études sur l'enseignement et l'apprentissage, Faculté des sciences de l'éducation, Université Laval, Québec, Québec, Canada, 8 Departement de Management, Faculté des Sciences de l'Administration, Université Laval, Québec, Canada, 9 Département de gestion, d'évaluation et de politique de santé, École de santé publique, Université de Montréal, Montréal, Québec, Canada

* ibeogo@uottawa.ca

**Data Availability Statement:** All relevant data are within the paper and its Supporting Information files.

## Abstract

### Background

Limited access to healthcare among children in sub-Saharan Africa (SSA) is a major cause of poor infant health indicators. Although many speculate that the private sector expansion has overwhelmingly reinforced health systems' utilization, little is known as to whether and where children are cared for when they are sick. This study investigated health-seeking behavior (HSB) among children from an urban area of Burkina Faso, with respect to disease severity and the type of provider versus children's characteristics.

### Methods

A cross-sectional population-based study was conducted in Ouagadougou, Burkina Faso using a two-stage sampling strategy. 1,098 households (2,411 children) data were collected. Generalized estimating equations (GEE) were used to analyze providers' choice for emergency, severe and non-severe conditions; sex-preference was further assessed with a $\chi 2$ test.

### Results

Thirty-six percent of children requiring emergency care sought private providers, as did 38% with severe conditions. Fifty-seven percent with non-severe conditions were self-medicated. A multivariable GEE indicated that University-educated household-heads would bring their

**Funding:** The first author received a fellowship from Taiwan-ICDF and a research grant from the International Health Program of National Yang Ming University. The publication fees were supported by the Faculty of Health Sciences, University of Ottawa. The funders had no role in study design, data collection and analysis, decision to publish, or preparation of the manuscript.

**Competing interests:** The authors have declared that no competing interests exist.

**Abbreviations:** AS, administrative sector; DH, district hospital; GEE, generalized estimating equations; FP, for-profit; HSB, health-seeking-behavior; LMIC, middle-income countries; NFP, not-for-profit; NISD, National Institute of Statistics and Demographics; PHC, primary care center; PPS, probability-proportional-to-size sampling; PSU, primary sampling unit; SSA, sub-Saharan Africa; TH, teaching hospital; SSU, secondary sampling unit; WHO, World Health Organization.

children to for-profit (instead of public) providers for emergency (OR = 3.51, 95%CI = 1.90; 6.48), severe (OR = 4.05, 95%CI: 2.24; 7.30), and non-severe (OR = 3.25, 95%CI = 1.25; 8.42) conditions. A similar pattern was observed for insured and formal jobholders. Children's sex, age and gender was not associated with neither the type of provider preference nor the assessed health condition.

## Conclusion

Private healthcare appeared to be crucial in the provision of care to children. The household head's socioeconomic status and insurance coverage significantly distinguished the choice of care provider. However, the phenomenon of son-preference was not found. These findings spotlighted children's HSB in Burkina Faso.

## Introduction

Since the endorsement of neoliberal policy in the 1990s by most of the African governments, a private health system has emerged in the healthcare market. Its presence is particularly observed in urban cities, leading to numerous therapeutic systems from diverse ownerships [1]. Elsewhere, in low-income countries, a number of studies have pinpointed that the private sector enhances the offer of healthcare and services, and contributes to boost competition, resulting in quality improvement [2–5].

The foundation of health services relies on the improvements in access and quality of care, mainly primary care which subsequently leads to better outcomes [6]. This includes access to all health services including curative and preventative care. Quality care is safe, effective, patient-centered, timely, efficient, and equitable [7]. For the Institute of Medicine, access to healthcare is correlated with children's overall health and wellbeing, including their physical and emotional growth [8]. However, in the context of sub-Saharan Africa (SSA), there is a paucity in the healthcare access literature, as few studies are devoted to this topic, particularly with regard to children's Healthcare-seeking behavior (HSB) in the present context.

With only 0.5 medical visits per capita per year [9], low utilization of health services in SSA is a major cause of poor child health indicators and continues to be at the top of researchers' and policy-makers' agenda. Recently in SSA [10], including Burkina Faso, the private health sector has grown and become very popular (Fig 1), reinforcing the traditional public health system. It is particularly concentrated in the largest cities [11]. Two systematic reviews on HSB in low- and middle-income countries (LMICs) [12, 13] illustrated how decision-making for healthcare encompasses all available options: seek care versus not seeking care, public versus private, and modern versus traditional. Whether children are brought for medical and where become critical policy questions.

Despite tremendous efforts channeled in Burkina Faso in recent decades to strengthen the health system, its utilization remains far behind the World Health Organization's (WHO) suggested level of 'one medical visit' per year per inhabitant [14].

Low health service utilization is a concern particularly for children and vulnerable subpopulations. Excess morbidity and mortality of children may largely result from failure of health systems to ensure adequate access to available treatment [15–17]. The broadest part of the literature targets more specifically the children under-five (U5) years old and further emphasizes a disease-specific HSB such as U5 febrile children (Adina et al., 2017), malaria [18], pneumonia

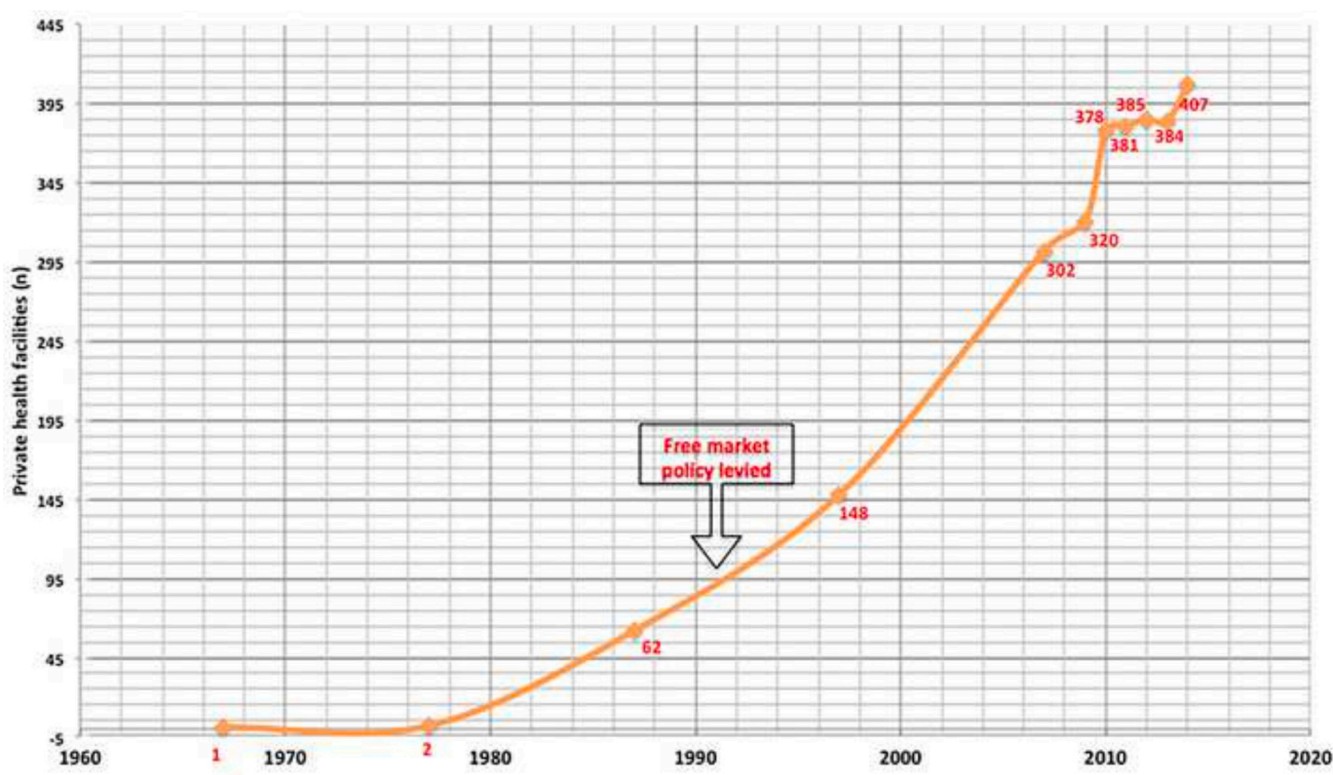

**Fig 1. Private health sector growth in Burkina Faso.** Sources: [42, 43].

[19, 20], to cite a few. Finally, very little literature has portrayed the under 15-year-old children HSB, in an urban context, and particularly in the current private health sector in Burkina Faso.

Healthcare Seeking Behavior is complex. The Andersen Emerging Behavioral Model [21] suggests healthcare utilization involves predisposing (gender, education...), enabling (occupation), and need variables (illness). The literature indicates that characteristics of household heads may be important determinants of demand for medical care [22, 23]. Predisposing factors might distinguish the choice between types of care and providers. Hjortsberg [24] also theorized the importance of socio-demographic factors in HSB. Of the aforementioned factors, the sex-related factor prevalently distinguished children HSB, namely son-preference in HSB. Several studies demonstrated a son-preference in healthcare decision-making in many LMICs [25], where boys are more likely to be reported ill than girls, and be brought to an external care provider given the illness [26, 27]. In their narrative review on HSB, Widayanti et al. [28] noted that Indonesian people sought help only when symptoms hindered their capacity to work, and not only patients decide; family, friends, and other community members are involved as well. The authors also contended that they refrained to seek public providers to avoid administrative complexities, long queuing, to cite some. In addition, more money would be spent with private providers if the child were a boy [26, 29].

As for enabling factors, studies contended the role of insurance and income in care utilization and the provider patronized. In Cassedy et al.'s three 2-year nationally representative panel study, children experiencing gaps in insurance coverage were less likely (4.5 odds) to access healthcare compared to children with continuous insurance [30]. In one study on community-based insurance, children from insured households were more likely to visit a formal

care provider than the non-insured [31]. In the same vein, insured children and those from the wealthiest income quintile prompted private provider utilization [26, 32] or were more likely to report illness episodes [26, 33].

Non-monetary factors were also shown to be associated with HSB. Pokhrel [33] and Pokhrel & Sauerborn [29] showed in their study that, individuals may go to a doctor and get into the healing process only once they perceive themselves to be sick or have trust in providers [34], or see who offers more acceptable services.

Hence, disease-related factors such as the type of disease or perceived severity are important determinants of HBS for child illnesses [35–40]. Finally, household factors, such as its size, matter in the decision to seek healthcare and where to seek it [25, 40].

The private sector is heterogeneous; however, few studies distinguish between formal versus informal, public versus for-profit (FP) versus not-for-profit (NFP) providers. Some categorize non-conventional providers (traditional healers) as private providers [26, 32, 41]. In the SSA health system, although traditional providers play an important role as a resort in health providing, they are not part of the formal health system, not plotted in the health pyramid.

A failure to disentangle facility ownership may however lead to misleading results and distort a robust comparability to inform policy for appropriate policy options. Despite an emerging private health system, our understanding of HSB for children's illnesses in urban areas is very limited. This population-based study investigated HSB among urban children with respect to disease severity and the type of provider and explored whether children's characteristics are associated with a specific provider preference. Understanding factors associated with health services' utilization for children' illnesses is essential for planners in Burkina Faso where one of the main policy goals of the government is to improve access and use of health services.

## Methods

### Setting

The study was conducted in Ouagadougou, Burkina Faso's capital city and main commercial hub. It is home to two million inhabitants, has the densest network of health facilities, including three out of the four national reference hospitals. In addition, the city has four district hospitals, an array of frontline facilities, and a headcount of approximately 60% of the country's private providers [11].

### Study population

The study population consists of the inhabitants of the Ouagadougou administrative municipality, distributed in 30 administrative sectors (AS). As detailed elsewhere (Beogo et al. 2014), in order to keep the urban perspective, the present study excluded the village attached to the Ouagadougou municipality. We also excluded the military barracks, the hospitals, and the commercial units. Finally, households where a household member resided in the city for less than six months were also excluded. We hypothesized that they might not be used to all the existing healthcare resources and therefore have not built a consistent health-seeking-behaviour associated with existing healthcare recourse. Households were selected through a two-stage cluster procedure. In the absence of a list of households, we applied the simplified general method for cluster-sample surveys developed by the WHO [44, 45]. To maximize the sample representativeness and strengthen the statistical power, we combined the cardinal point system to determine the clusters (streets). Based on the city map, South, North, West, and East were drawn as the entry points order for the 30 ASs. We then randomly drew (without replacement) the streets (primary sampling unit [PSU]) in each of the individual ASs. From each selected street, the right-side households (secondary sampling unit [SSU]) were drawn applying a skip

interval. The selection was continued —left side households and other cardinal points— until the targeted number was attained. The survey randomly selected one household in common plots and excluded the commercial streets. Furthermore, empty plots were excluded from the skip interval count. The probability-proportional-to-size sampling (PPS) was applied. The number surveyed was based on the sample size allocated to each selected street; the headcount of households was obtained from the Burkina Faso National Institute of Statistics and Demographics (NISD).

## Sample size and sampling

Formula for estimating single proportion as employed elsewhere [46] was implemented to calculate the sample size, and its consistency compared with other previous investigations of its kind, carried out in LMICs, including Burkina Faso [35, 47, 48], Nigeria [49], Eritrea [50], and Pakistan [51].

$$n = \frac{\beta_{\alpha/2}^2 \pi (1 - \pi)}{\delta^2}$$

Let $\delta = 0.025$, $\beta = 1.96$, $\alpha = 0.05$. Due to the limited literature on the incidence of the health system utilization, we supposed $\pi = 50\%$. This resulted on a size of 1537 households that we extended to 1600 in order to take into account incomplete questionnaires.

A household was considered a group of people whose food was prepared by the same person [44]. Consenting households with at least one child were selected. Similar to studies conducted in East Africa [50, 52], West Africa [53] including Burkina Faso [31], and elsewhere [32, 54], children were defined as household members who were younger than 15.

## Data collection and instrument

We used a structured questionnaire adapted from the multiple indicator cluster surveys' questionnaires of the NISD, and similar surveys conducted elsewhere. The questionnaire records household level information (household size, health payment scheme. . .) and individual characteristics (age, sex, filiation. . .).

Data were collected after the questionnaire content validity was assessed. It was then translated forward and backward (English-French) and finally pilot-tested in 32 households from four ASs. In each household, the head and spouse (if any) were interviewed separately face-to-face. In case of conflicting information, the spouse's response was retained, assuming she would be more aware of any illness events occurring in the household. The questions posed were straightforward, looking for the usual choice of care provider that they take their children to when encountering an emergency, or severe, and non-severe conditions. An example of a framed question was: "To which source of care do you bring (a specific child) when she/he experiences an emergency disease/injury?" An emergency condition was described as a condition that the caregiver perceives that may result in a fatality in the absence of urgent intervention. The examples of common symptoms included loss of consciousness, convulsions, open fracture, and eye rolling. For a severe condition, there was not a life-threatening perception, though an intervention was needed to avoid aggravation. Shortness of breath, high fever, cough with blood, sunken fontanel or sunken eyes, severe abdominal pain, and injuries caused by an accident were some illustrative symptoms. Finally, non-severe conditions were defined as the absence of perception of vital danger but compromised daily duties if medical care was not established. An array of symptoms was listed: headaches, stomach aches, fevers, common cold or coughs.

Six trained enumerators collected data. Everyone was given a booklet detailing the field-work and a hotline was made available for any matter they could encounter. The principal investigator (IB) supervised the streets selection (12%) and repeated interviews (2%) using a shortened version of the questionnaire with unalterable variables. Data were entered using the Census and Survey Processing package (CS-Pro), Version 4.0.

## Study variables

The primary outcome variable was the type of care provider participants might resort to: formal and informal. Formal providers consisted of public and private providers. The first is composed of frontline facilities (primary care center [PHC]), district hospital (DH), and teaching hospital (TH). The private provider included FP and NFP. For the purpose of this paper, self-medication and traditional healers were aggregated into the single category "informal provider". Explanatory variables comprised both child and household characteristics: age, gender, insurance status, and their relationship with household head. Household characteristics included its size, household head's education, marital status, and occupation (Table 1). Furthermore, a set of 10 commonly identified reasons determining provider choice was also assessed.

## Statistical analysis

A descriptive analysis was carried out. Pearson's $\chi^2$ test was fitted to examine sex differences in providers' choice. The correlated nature of data led to implement a generalized estimating equation (GEE) as suggested elsewhere [55, 56]. For this complexity, and in order determine the potential role played by the Administrative Sector (level1) and household (level2), we ran a multilevel analysis. Only the implementation of Poisson regression resulted in converging, possibly due to effects of the data complex structure or overparameterization [57] (See S1 Table). The robust variance estimates by the GEE approach took into account the potential within the household correlation; providers' choice was regressed with individual (children) and household level variables, and thus gave better estimate of the standard error of the effect.

**Table 1. Variables definition.**

| *Dependent variables* | |
|---|---|
| Type of providers | 1: conventional, 2: otherwise |
| Type of conventional providers | 1: for-profit, 2: not-for-profit, 3: public |
| *Explanatory variables* | |
| Household size | 1: 1 kids, 1: 2–3 kids, 1: >3 kids |
| Insurance | 1: insured, 2: not insured |
| Age, year | Children [1: <1, 2: 1–4, 2: 5–14], household head [1: 15–24, 2: 25–44, 3: 45–64, 4:> = 65] |
| Sex | 1: female, 2: male |
| Filiation | 1: son\|daughter\| grandson\|granddaughter, 2: other children (brother, nephew, cousin…) |
| Gender | 1: female, 2: male |
| Marital status | 1: married (formal+ free union), 2: otherwise |
| Education level | 1: university 2: secondary, 3: primary, 4: no formal education |
| Employment status | 1: government job, 2: parapublic & formal private 3: informal private job, 4: not in labor |

We also tested the potential role of the administrative sector and the household levels through a sensitivity analysis (multilevel analysis). Analyses were run using SPSS Version 21 and SAS version 14.3, for the multilevel analysis.

### Ethics approval

We interviewed only household heads and their spouse. They answered the questions for themselves and on the behalf of all other household members, children included. The informed consent statement, presented on the front page of the questionnaire, was read and explained to participants. To maintain a friendly environment of communication, verbal consent was sought instead of written consent. Also, a considerable proportion of interviewees in the target population may not have been literate; second, in prior field experiences, concerns were raised that a written signature would compromise the anonymity of the interview and compromise interviewees' trust. Finally, the same strategy is also used in Demographic and Health Surveys (DHSs). Although the respondents were not asked to sign the form, it was either read to them or they were allowed to read it themselves and ticked "yes" or "no" accordingly. When the response was "no," the interviewers did not attempt to convince them otherwise. Interviewers were asked to politely apologize and leave. The targeted respondents were clearly informed of the voluntary nature of their participation and could decline their consent at any time. An ethical clearance to conduct the study, including the consenting procedure, was granted by the Burkina Faso National Ethics Committee for Research (#2011-11-82). In addition, the Town Council of Ouagadougou granted an administrative approval. All data used were anonymized to protect the respondents' privacy throughout the process.

### Results

Of the 1600 households surveyed, 1120 had at least one child. Of them, twenty-two were excluded, two residing in the city for less than 6 months, and the rest (n = 20) failed to complete the interview process. The remaining 1098 households led to a sample of 2,411 children. Respectively 74, 82, and 80 children with missing information when reporting for emergency, severe, and non-severe conditions were removed from the analyses. Finally, the analytical data for emergency, severe and non-severe conditions were 2337, 2329, and 2331, respectively (Fig 2). Almost universally, 96.6% (n = 2257) and 97.9% (n = 2280) of children were brought to formal care providers for emergency and severe conditions, respectively. The majority (61.1%, n = 1428) of children's caregivers sought care from a public provider for an emergent pediatric condition, while 35.5% (n = 886) chose private providers. Among private provider seekers, slightly more than half went to NFP providers (51.4%). For a severe condition, 59.8% (n = 1394) preferred public providers and 38.0% (n = 886) private providers. Within the latter, 52.8% (n = 468) patronized FP. On the contrary, the great majority (56.7%, n = 1322) dealt with a non-severe condition with an informal care providers.

Table 2 exhibits participants' characteristics. Most of the surveyed children were girls (53.5%), 30% were younger than five-years old, and 3.6% held insurance coverage. In terms of household characteristics, 84.0% of the households had one child or more, and 85.0% were headed by a male. About 45% of the household heads had at least a secondary education level and 30% had a formal job either in government or in the formal private sector.

Table 3 presents the distribution of provider choice among people who indicated they would seek formal care. In most cases, regardless of their gender, age and filiation, children were brought to a public provider. Within private provider seekers, NFP providers were slightly more visited across conditions for children younger than one-year old: 24.6% (n = 31), 21.9% (n = 28), and 23.0% (n = 14), respectively for emergency, severe, and non-severe

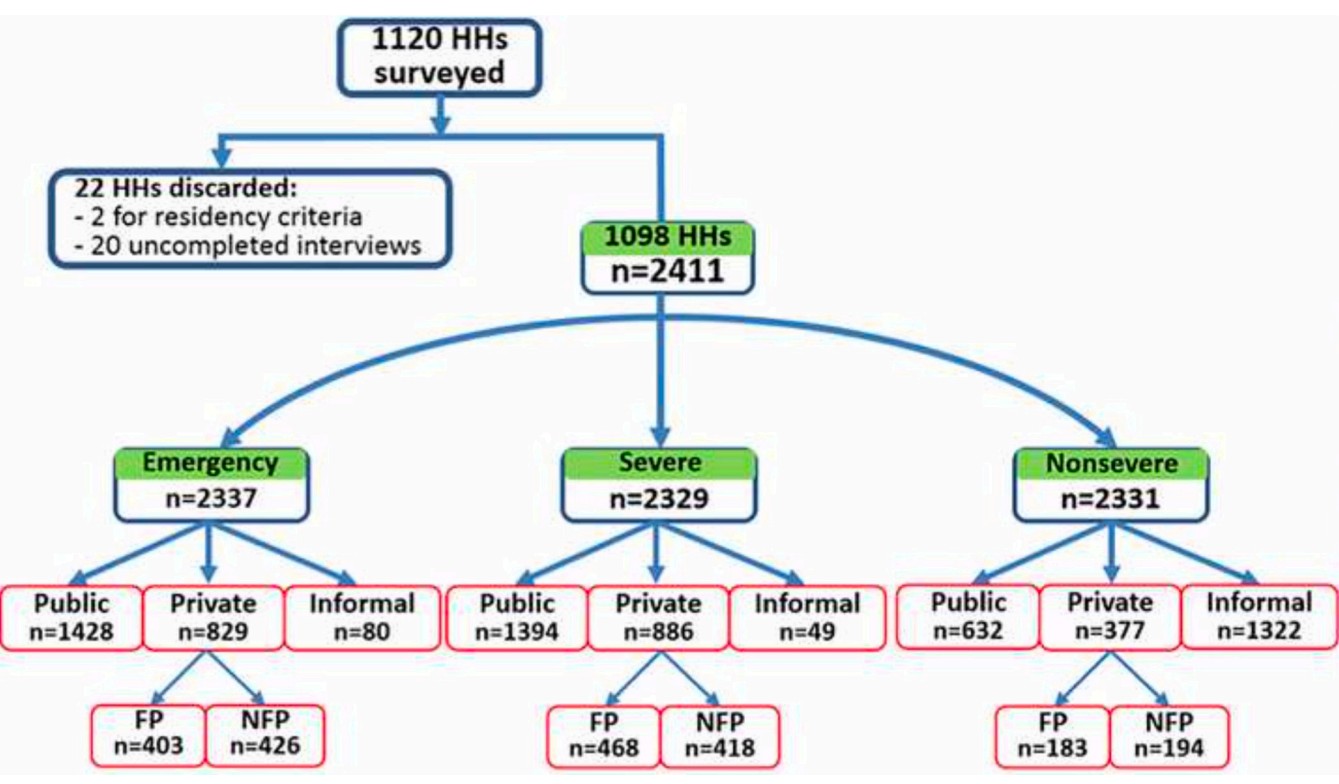

**Fig 2. Healthcare providers' utilization.**

conditions. Insured children markedly chose FP providers. They were, hence, 46.7% (n = 36), 57.5% (n = 46) and 58.7% (n = 27) to patronize FP providers for emergency, severe, and non-severe conditions, respectively. A similar pattern was noted for children whose household head had attended a university level. About half of the household heads holding a good job position—government, parapublic, and formal private—preferred to be attended to by FP providers. That proportion was far greater for parapublic and formal private workers: 28.3% (n = 85), 33.5% (n = 101), 28.6% (n = 44), respectively for emergency, severe, and non-severe conditions. By contrast, for both household heads not employed and those with no formal education, opted for public sector providers for any encountered condition.

### Children's characteristics in the choice of provider

Table 4 shows age and gender preferences. For all providers, irrespective of children's gender and age bracket, a large proportion went to the formal sector across the three medical conditions. A very small proportion sought informal providers. No significant difference in seeking care between public, private, and informal providers was found (Pearson's chi-square test). A further stratified analysis on formal providers (Table 4, bottom) did not show a significant difference between girls and boys.

### Multivariate analysis

Table 5 shows the multivariable GEE of those who sought a formal provider. The analysis, conditional on the individual choosing FP versus public and NFP versus public, integrated in the models the household head's characteristics, as infant HSB relies on his/her decision. No significant association was laid out for children's age, gender, and filiation for all conditions.

**Table 2. Sample distribution by characteristics and types of health condition.**

| | Emergency (n = 2337) | | Severe (n = 2329) | | Non-severe (n = 2331) | |
|---|---|---|---|---|---|---|
| | n | % | n | % | n | % |
| **Individual characteristics** | | | | | | |
| **Age**, year | | | | | | |
| <1 | 127 | 5.4 | 129 | 5.5 | 125 | 5.4 |
| 1–4 | 569 | 24.4 | 567 | 24.4 | 564 | 24.2 |
| 5–14 | 1641 | 70.2 | 1633 | 70.1 | 1642 | 70.4 |
| **Gender** | | | | | | |
| Girl | 1,251 | 53.5 | 1242 | 53.3 | 1248 | 53.5 |
| Boy | 1,086 | 46.5 | 1087 | 46.7 | 1083 | 46.5 |
| **Filiation** | | | | | | |
| Son\|Daughter\|grd Son\|Daughter | 2198 | 94.1 | 2190 | 94.0 | 2190 | 94.0 |
| Other children | 139 | 5.9 | 139 | 6.0 | 141 | 6.0 |
| **Insurance** | | | | | | |
| Insured | 86 | 3.7 | 86 | 3.7 | 84 | 3.6 |
| Not insured | 2251 | 96.3 | 2243 | 96.3 | 2247 | 96.4 |
| **Household Characteristics** | | | | | | |
| **Household size** | | | | | | |
| 1 child | 373 | 16.0 | 373 | 16.0 | 374 | 16.0 |
| 2–3 children | 1307 | 55.9 | 1307 | 56.1 | 1304 | 55.9 |
| >3 children | 657 | 28.1 | 649 | 27.9 | 653 | 28.0 |
| **Headship's age**, year | | | | | | |
| 15–24 | 27 | 1.2 | 27 | 1.2 | 28 | 1.2 |
| 25–44 | 1203 | 51.5 | 1194 | 51.5 | 1198 | 51.4 |
| 45–64 | 893 | 38.2 | 894 | 38.5 | 890 | 38.2 |
| > = 65 | 205 | 8.8 | 205 | 8.8 | 205 | 8.8 |
| **Headship's gender** | | | | | | |
| Men | 336 | 14.4 | 332 | 14.3 | 328 | 14.1 |
| Women | 1995 | 85.6 | 1991 | 85.7 | 1996 | 85.9 |
| **Headship's marital status** | | | | | | |
| In union | 2121 | 91.0 | 2117 | 91.1 | 2118 | 91.0 |
| Otherwise | 210 | 9.0 | 206 | 8.9 | 206 | 9.0 |
| **Headship's education level** | | | | | | |
| Primary | 345 | 14.9 | 342 | 14.8 | 345 | 14.9 |
| Secondary | 750 | 32.4 | 741 | 32.1 | 748 | 32.4 |
| University | 314 | 13.5 | 314 | 13.6 | 314 | 13.6 |
| No formal education | 908 | 39.2 | 912 | 39.5 | 905 | 39.1 |
| **Headship's occupation** | | | | | | |
| Government job | 393 | 16.9 | 393 | 17.0 | 394 | 17.0 |
| Parapublic & formal private | 319 | 13.7 | 312 | 13.5 | 319 | 13.8 |
| Informal private | 1160 | 50.0 | 1162 | 50.2 | 1159 | 50.1 |
| Not in labor | 449 | 19.4 | 446 | 19.3 | 442 | 19.1 |

Insurance however appeared to significantly predict the choice of FP providers for a severe condition (Odd Ratio [OR] = 1.99, 95% Confidence Interval [CI] = 1.16; 3.40) and a non-severe one (OR = 3.29, 95% CI = 1.37; 7.87). The insured were 37% more likely to choose FP providers for an emergency, though the CI exhibited a marginal significance. Apart from insurance, the parents'—herein, the household head's—occupation and education were also

**Table 3. Distribution of factors associated with the providers' (FP, NFP & public) choice by health condition.**

| | Emergency (n = 2257) | | | | Severe (n = 2280) | | | | Non-severe (n = 1009) | | | |
|---|---|---|---|---|---|---|---|---|---|---|---|---|
| | n | FP (%) | NFP (%) | Pub (%) | n | FP (%) | NFP (%) | Pub (%) | n | FP (%) | NFP (%) | Pub (%) |
| **Individual characteristics** | | | | | | | | | | | | |
| **Age**, year | | | | | | | | | | | | |
| <1 | 126 | 14.3 | 24.6 | 61.1 | 128 | 20.3 | 21.9 | 57.8 | 61 | 21.3 | 23.0 | 55.7 |
| 1–4 | 546 | 18.5 | 18.7 | 62.8 | 557 | 22.1 | 18.8 | 59.1 | 249 | 18.5 | 17.3 | 64.2 |
| 5–14 | 1585 | 17.9 | 18.5 | 63.6 | 1595 | 20.0 | 17.9 | 62.1 | 699 | 17.7 | 19.6 | 62.7 |
| **Gender** | | | | | | | | | | | | |
| Girl | 1208 | 18.0 | 19.0 | 63.0 | 1218 | 20.8 | 18.6 | 60.7 | 530 | 17.2 | 18.1 | 64.7 |
| Boy | 1049 | 17.6 | 18.8 | 63.6 | 1062 | 20.2 | 18.1 | 61.7 | 479 | 19.2 | 20.5 | 60.3 |
| **Filiation** | | | | | | | | | | | | |
| HHH's son-daughter | 2124 | 18.1 | 19.0 | 62.9 | 2143 | 20.3 | 18.4 | 61.3 | 957 | 18.4 | 18.8 | 62.8 |
| Other children | 133 | 13.5 | 17.3 | 69.2 | 137 | 24.1 | 17.5 | 58.4 | 52 | 13.5 | 26.9 | 59.6 |
| **Insurance** | | | | | | | | | | | | |
| Insured | 77 | 46.7 | 20.8 | 32.5 | 80 | 57.5 | 21.3 | 21.2 | 46 | 58.7 | 23.9 | 17.4 |
| Not insured | 2180 | 16.8 | 18.8 | 64.4 | 2200 | 19.2 | 18.2 | 62.6 | 963 | 16.2 | 19.0 | 64.8 |
| **Household Characteristics** | | | | | | | | | | | | |
| **Household size** | | | | | | | | | | | | |
| 1 child | 365 | 18.6 | 22.2 | 59.2 | 368 | 22.6 | 20.1 | 57.3 | 180 | 17.3 | 19.4 | 63.3 |
| 2–3 children | 1248 | 17.8 | 18.9 | 63.3 | 1276 | 21.0 | 19.5 | 59.5 | 550 | 20.2 | 19.1 | 60.7 |
| >3 children | 644 | 17.5 | 16.9 | 65.5 | 636 | 18.4 | 14.9 | 66.7 | 279 | 14.7 | 19.4 | 65.9 |
| **Headship's age**, year | | | | | | | | | | | | |
| 15–24 | 27 | 0.0 | 25.9 | 74.1 | 27 | 11.1 | 18.5 | 70.4 | 15 | 0.0 | 13.3 | 86.7 |
| 15–44 | 1161 | 17.3 | 19.3 | 63.4 | 1173 | 22.7 | 20.3 | 57.0 | 531 | 20.2 | 17.7 | 62.1 |
| 45–64 | 857 | 20.5 | 16.3 | 63.1 | 868 | 20.4 | 15.1 | 64.5 | 367 | 17.4 | 17.5 | 65.1 |
| > = 65 | 205 | 11.2 | 24.9 | 63.9 | 205 | 9.3 | 19.5 | 71.2 | 90 | 8.9 | 35.6 | 55.5 |
| **Headship's gender** | | | | | | | | | | | | |
| Men | 323 | 14.5 | 25.1 | 60.4 | 324 | 17.9 | 23.8 | 58.3 | 93 | 6.4 | 23.7 | 69.9 |
| Women | 1930 | 18.4 | 17.7 | 63.9 | 1952 | 21.0 | 17.3 | 61.7 | 913 | 19.3 | 18.6 | 62.1 |
| **Headship's Marital status** | | | | | | | | | | | | |
| In union | 2055 | 18.4 | 18.4 | 63.2 | 2077 | 21.4 | 18.2 | 60.4 | 939 | 18.7 | 18.9 | 62.4 |
| Otherwise | 198 | 12.6 | 21.7 | 64.7 | 199 | 12.1 | 17.6 | 70.3 | 67 | 9.0 | 22.4 | 68.7 |
| **Headship's education** | | | | | | | | | | | | |
| No formal education | 884 | 11.3 | 21.2 | 67.5 | 900 | 13.7 | 19.7 | 66.6 | 361 | 8.9 | 26.6 | 64.5 |
| Primary | 336 | 18.7 | 16.7 | 64.6 | 335 | 18.8 | 19.4 | 61.8 | 113 | 13.3 | 12.4 | 74.3 |
| Secondary | 722 | 18.7 | 17.7 | 63.6 | 727 | 20.5 | 15.8 | 63.7 | 340 | 20.9 | 16.5 | 62.6 |
| University | 297 | 35.0 | 16.8 | 48.2 | 300 | 43.0 | 18.7 | 38.3 | 184 | 34.8 | 13.6 | 51.6 |
| **Headship's employment** | | | | | | | | | | | | |
| Public | 375 | 21.1 | 16.0 | 62.9 | 382 | 26.4 | 17.3 | 56.3 | 231 | 23.8 | 10.8 | 65.4 |
| Parapub & formal private | 300 | 28.3 | 18.3 | 53.3 | 301 | 33.5 | 19.3 | 47.2 | 154 | 28.6 | 21.4 | 50.0 |
| Informal private | 1129 | 14.9 | 20.4 | 64.7 | 1146 | 17.3 | 18.3 | 64.4 | 441 | 12.0 | 23.6 | 64.4 |
| Not in labor | 439 | 16.2 | 17.5 | 63.2 | 437 | 15.6 | 18.3 | 66.1 | 177 | 16.9 | 17.0 | 66.1 |

observed to be significantly associated with visiting FP providers. When holding other factors and considering the clustering effects, household heads with a university level were significantly more likely to choose a FP (instead of public) provider (OR = 3.51, 95% CI = 1.90; 6.48) for an emergency, (OR = 4.05, 95% CI = 2.24; 7.30) for severe, and (OR = 3.25, 95% CI = 1.253; 8.42) for non-severe conditions. Similarly, a good jobholder (parapublic and

**Table 4. Pearson's χ² test of age and gender (girls vs. boys) stratification in the choice of providers by health condition among urban children.**

| All providers | <1 year | | | | | 1–4 year | | | | | 5–14 year | | | | | All ages combined | | | | |
|---|---|---|---|---|---|---|---|---|---|---|---|---|---|---|---|---|---|---|---|---|
| | n | Prvt (%) | Pub (%) | Infl (%) | p | n | Prvt (%) | Pub (%) | Infl (%) | p | n | Prvt (%) | Pub (%) | Infl (%) | p | n | Prvt (%) | Pub (%) | Infl (%) | p |
| Emergency | | | | | | | | | | | | | | | | | | | | |
| Girl | 65 | 35.4 | 63.1 | 1.5 | 0.49 | 67 | 38.8 | 59.7 | 1.5 | 0.50 | 63 | 19.0 | 27.0 | 54.0 | 0.75 | 1251 | 35.7 | 60.8 | 3.4 | 0.96 |
| Boy | 62 | 41.9 | 58.1 | 0.0 | | 62 | 45.2 | 54.8 | 0.0 | | 62 | 24.2 | 27.4 | 48.4 | | 1086 | 35.2 | 61.4 | 3.4 | |
| Severe | | | | | | | | | | | | | | | | | | | | |
| Girl | 298 | 36.6 | 59.1 | 4.4 | 0.80 | 295 | 40.0 | 58.0 | 2.0 | 0.88 | 297 | 14.5 | 27.3 | 58.2 | 0.45 | 1242 | 38.6 | 59.5 | 1.9 | 0.73 |
| Boy | 271 | 34.7 | 61.6 | 3.7 | | 272 | 40.4 | 58.1 | 1.5 | | 267 | 17.2 | 29.6 | 53.2 | | 1087 | 37.4 | 60.3 | 2.3 | |
| Non-severe | | | | | | | | | | | | | | | | | | | | |
| Girl | 888 | 35.5 | 61.3 | 3.3 | 0.91 | 880 | 38.1 | 60.0 | 1.9 | 0.22 | 888 | 14.9 | 27.6 | 57.5 | 0.39 | 1248 | 15.0 | 27.5 | 57.5 | 0.25 |
| Boy | 753 | 34.8 | 61.6 | 3.6 | | 753 | 35.7 | 61.5 | 2.8 | | 754 | 17.1 | 25.6 | 57.3 | | 1083 | 17.5 | 26.7 | 55.8 | |
| **Formal providers** | n | FP (%) | NFP (%) | Pub (%) | p | n | FP (%) | NFP (%) | Pub (%) | p | n | FP (%) | NFP (%) | Pub (%) | p | n | FP (%) | NFP (%) | Pub (%) | p |
| Emergency | | | | | | | | | | | | | | | | | | | | |
| Girl | 64 | 14.1 | 21.9 | 64.1 | 0.74 | 285 | 18.9 | 19.3 | 61.8 | 0.86 | 859 | 18.0 | 18.6 | 63.3 | 0.97 | 1208 | 18.0 | 19.0 | 63.0 | 0.96 |
| Boy | 62 | 14.5 | 27.4 | 58.1 | | 261 | 18.0 | 18.0 | 64.0 | | 726 | 17.8 | 18.3 | 63.9 | | 1049 | 17.6 | 18.8 | 63.6 | |
| Severe | | | | | | | | | | | | | | | | | | | | |
| Girl | 66 | 19.7 | 19.7 | 60.6 | 0.77 | 289 | 21.5 | 19.4 | 59.2 | 0.90 | 863 | 20.6 | 18.2 | 61.2 | 0.68 | 1218 | 20.8 | 18.6 | 60.7 | 0.89 |
| Boy | 62 | 21.0 | 24.2 | 54.8 | | 268 | 22.8 | 18.3 | 59.0 | | 732 | 19.3 | 17.5 | 63.3 | | 1062 | 20.2 | 18.1 | 61.7 | |
| Non-severe | | | | | | | | | | | | | | | | | | | | |
| Girl | 29 | 24.1 | 17.2 | 58.6 | 0.58 | 124 | 16.9 | 17.7 | 65.3 | 0.82 | 377 | 16.7 | 18.3 | 65.0 | 0.38 | 530 | 17.2 | 18.1 | 64.7 | 0.36 |
| Boy | 32 | 18.8 | 28.1 | 53.1 | | 125 | 20.0 | 16.8 | 63.2 | | 322 | 18.9 | 21.1 | 59.9 | | 479 | 19.2 | 20.5 | 60.3 | |

**Abbreviation**: FP, for profit; NFP, not-for-profit; Prvt, private; Pub, public; Infl, informal; P, p-value

formal private) that insures financial solvency was inclined to present his children to a FP provider when encountering an emergency (OR = 1.86, 95% CI = 1.03; 3.35) or a severe condition (OR = 1.86, 95% CI = 1.02; 3.39).

As shown by Pearson's χ² test, the multivariable GEE also did not reveal any gender preference in the type of provider choice: this, neither for boys nor girls nor in the headship's sex for all three medical conditions.

Overall, contrary to the differences noted between FP and public providers, no significant differences in individual characteristics as well as in household ones were observed between those who chose public versus NFP providers. This observation applied to the three assessed medical conditions.

In order to determine a potential role played by the AS (level1) and household (level2), a multilevel sensibility analysis test was tried, but failed to converge − possible effects of the data complex structure or overparameterized (Bates, 2015) − this, even considering Poisson distribution. Only, the AS as level of analysis converged for emergency and not severe conditions (see S1 Table), severe condition did not. Therefore, no complete comparative feature could be done with the GEE analysis.

Table 6 presents the proximity to the providers as the top reason leading to the choice of the three defined groups of providers. The competence of the provider was steadfast in second position for FP and NFP providers across conditions, while it was the cheapness of services for the public provider for emergency and non-severe conditions. The additional differences regard the constant choice of FP provider for their promptness in servicing as the third leading reason. This distinguishes other providers' choice. Finally, the 24-hour service stood for the

**Table 5. Multivariate GEE of the choice between the formal providers by urban children by health condition.**

| Individual characteristics | Emergency | | | | Severe | | | | Non-severe | | | |
|---|---|---|---|---|---|---|---|---|---|---|---|---|
| | FP vs. Public | | NFP vs. Public | | FP vs. Public | | NFP vs. Public | | FP vs. Public | | NFP vs. Public | |
| | OR | 95%CI | OR | 95%CI | OR | 95%CI | OR | 95%CI | OR | 95%CI | OR | 95%CI |
| Age, year | | | | | | | | | | | | |
| 5–14 | 1 | | 1 | | 1 | | 1 | | 1 | | 1 | |
| <1 | 1.01 | 0.97; 1.05 | 1.03 | 1.00; 1.07 | 1.02 | 0.96; 1.08 | 1.01 | 0.99; 1.02 | 1.04 | 0.84; 1.29 | 1.02 | 1.00; 1.04 |
| 1–4 | 1.00 | 0.98; 1.02 | 1.00 | 0.99; 1.01 | 1.01 | 0.99; 1.04 | 1.00 | 1.00; 1.01 | 0.90 | 0.72; 1.04 | 1.00 | 0.99; 1.01 |
| Gender | | | | | | | | | | | | |
| Boy | 1 | | 1 | | 1 | | 1 | | 1 | | 1 | |
| Girl | 1.00 | 0.98; 1.01 | 1.00 | 0.99; 1.01 | 1.00 | 0.97; 1.03 | 1.00 | 1.00; 1.01 | 0.91 | 0.80; 1.04 | 1.00 | 1.00; 1.00 |
| Filiation | | | | | | | | | | | | |
| Other children | 1 | | | | | | | | | | | |
| Headship's Son-D\|grdSon-D | 1.03 | 0.90; 1.01 | 1.01 | 0.97; 1.10 | 0.95 | 0.87; 1.03 | 1.00 | 0.99; 1.02 | 1.20 | 0.80; 1.80 | 0.99 | 0.95; 1.02 |
| Insurance | | | | | | | | | | | | |
| Not Insured | 1 | | 1 | | 1 | | 1 | | 1 | | 1 | |
| Insured | 1.37 | 0.91; 2.07 | 1.37 | 0.85; 2.23 | 1.99 | 1.16; 3.40 | 1.27 | 0.92; 1.74 | 3.29 | 1.37; 7.87 | 1.17 | 0.94; 1.46 |
| Household Characteristics | | | | | | | | | | | | |
| Household size | | | | | | | | | | | | |
| >3 children | 1 | | 1 | | 1 | | 1 | | 1 | | 1 | |
| 1 child | 1.04 | 0.61; 1.80 | 1.54 | 0.90; 2.63 | 1.20 | 0.72; 2.01 | 1.41 | 0.80; 2.48 | 1.05 | 0.49; 2.47 | 1.24 | 0.59; 2.61 |
| 2–3 children | 0.99 | 0.59; 1.65 | 1.23 | 0.73; 2.05 | 1.09 | 0.66; 1.79 | 1.39 | 0.81; 2.39 | 1.33 | 0.59; 3.02 | 1.18 | 0.58; 2.42 |
| Headship's head age, year | 1.01 | 1.00; 1.03 | 1.00 | 0.98; 1.01 | 0.99 | 0.98; 1.01 | 0.99 | 0.98; 1.00 | 1.00 | 0.97; 1.03 | 1.02 | 1.00; 1.04 |
| Headship's gender | | | | | | | | | | | | |
| Men | 1 | | 1 | | 1 | | 1 | | 1 | | 1 | |
| Women | 1.01 | 0.50; 2.03 | 1.70 | 0.98; 2.95 | 1.55 | 0.80; 3.01 | 2.60 | 1.52; 4.46 | 0.35 | 0.10; 1.29 | 1.09 | 0.40; 2.95 |
| Headship's marital status | | | | | | | | | | | | |
| In union | 1 | | 1 | | 1 | | 1 | | 1 | | 1 | |
| Otherwise | 0.92 | 0.43; 2.00 | 1.46 | 0.74; 2.87 | 1.77 | 0.80; 3.87 | 2.44 | 1.25; 4.77 | 0.82 | 0.18; 3.64 | 1.10 | 0.33; 3.67 |
| Headship's education | | | | | | | | | | | | |
| No formal education | 1 | | 1 | | 1 | | 1 | | 1 | | 1 | |
| Primary | 2.00 | 1.16; 3.46 | 0.95 | 0.58; 1.57 | 1.80 | 1.06; 3.05 | 1.13 | 0.69; 1.86 | 1.55 | 0.618; 3.91 | 0.53 | 0.22; 1.27 |
| Secondary | 1.73 | 1.03; 2.89 | 0.98 | 0.63; 1.51 | 1.38 | 0.86; 2.21 | 0.75 | 0.48; 1.16 | 2.15 | 0.928; 5.00 | 0.89 | 0.45; 1.76 |
| University | 3.51 | 1.90; 6.48 | 1.31 | 0.73; 2.38 | 4.05 | 2.24; 7.30 | 1.47 | 0.80; 2.68 | 3.25 | 1.253; 8.42 | 0.96 | 0.40; 2.30 |
| Headship's employment | | | | | | | | | | | | |
| Not in labor | 1 | | 1 | | 1 | | 1 | | 1 | | 1 | |
| Government job | 1.19 | 0.66; 2.16 | 1.28 | 0.70; 2.35 | 1.35 | 0.742; 2.4 | 1.26 | 0.69; 2.29 | 0.54 | 0.21; 1.38 | 1.21 | 0.44; 3.36 |
| Parapub & formal private | 1.86 | 1.03; 3.35 | 1.74 | 0.95; 3.17 | 1.86 | 1.02; 3.39 | 1.59 | 0.87; 2.90 | 0.77 | 0.28; 2.12 | 2.13 | 0.77; 5.89 |
| Informal private | 1.08 | 0.65; 1.79 | 1.52 | 0.95; 2.43 | 1.08 | 0.65; 1.77 | 1.00 | 0.63; 1.58 | 0.60 | 0.26; 1.39 | 1.97 | 0.91; 4.27 |

third reason for going to a NFP provider—for emergency and severe conditions—while competence ranked third for seeking the public sector of care for emergency and not-severe conditions.

## Discussion

This large-scale population-based study highlighted important patterns of HSB for children in Ouagadougou, an urban city with an emerging private care sector.

First, for emergency or severe conditions, children were predominantly brought to formal care providers, whereas the majority of children were self-treated at home for non-severe

**Table 6. Reasons prompting different formal providers' (FP, NFP, Public) choice.**

| Reasons | Emergency | | | Severe | | | Non-severe | | |
|---|---|---|---|---|---|---|---|---|---|
| | FP (n:401) | NFP (n:425) | Pub (n:1423) | FP (n: 468) | NFP (n:418) | Pub (n:1390) | FP (n:180) | NFP (n:190) | Pub (n:630) |
| H24 services (%) | 3.7 | 6.4 | 4.2 | 6.8 | 12.2 | 3.8 | 0.6 | 1.1 | 1.3 |
| Proximity (%) | 48.4 | 64.5 | 73.4 | 35.0 | 46.9 | 47.9 | 55.6 | 64.2 | 78.3 |
| Promptness (%) | 11.0 | 5.6 | 3.9 | 17.7 | 9.8 | 6.5 | 7.2 | 3.2 | 0.8 |
| Competence (%) | 31.2 | 15.3 | 7.4 | 32.1 | 23.0 | 28.1 | 23.3 | 14.2 | 3.0 |
| Good drugs (%) | 0.0 | 0.0 | 0.6 | 2.1 | 0.2 | 1.4 | 0.0 | 0.0 | 0.3 |
| Good equipment (%) | 0.0 | 0.0 | 0.2 | 1.1 | 1.2 | 1.9 | 0.0 | 0.0 | 0.0 |
| Prior satisfaction (%) | 2.2 | 5.2 | 1.2 | 2.4 | 3.1 | 1.3 | 2.2 | 6.3 | 0.5 |
| Cheapness (%) | 1.0 | 1.2 | 8.2 | 0.9 | 2.4 | 7.9 | 2.2 | 6.8 | 14.1 |
| Connection (%) | 2.5 | 1.9 | 0.8 | 1.9 | 1.2 | 1.2 | 4.4 | 4.2 | 1.3 |
| Other reasons* (%) | 0.0 | 0.0 | 0.0 | 0.0 | 0.0 | 0.0 | 4.4 | 0.0 | 0.5 |

FP, for-profit; NFP, not-for-profit; Pub, public

*Accept credit, often offer medication for free, offer emergency care on credit. . .

conditions. Disease severity did play a significant role in determining whether children would seek care from a formal or informal provider. Furthermore, among formal care providers, public providers remained the main source of care for the majority of children, regardless of the severity of the medical condition. Convenient location of the first-line public facilities and certainly the cheapness of services provided are plausible reasons why children may be brought to public providers. Yet, PHC remains the foremost interface for the government public health program, an essential point of delivery of prenatal care and immunization. Hence, possibly due to the integrated array package of various services offered in PHC, women, typically the main caregivers of children may be more acquainted with these PHCs and have trust in their competence.

Apart from seeking a formal care provider, more than one-third of the children are brought to private providers. Compared to the pattern observed in a study two decades ago in the same study setting [58], it clearly appears that private provision of care and services have critically augmented. We argue that such an important proportion resorting to the private sector, −35.5%, emergency; 38.0%, severe−, informs that people successfully adjust to the new health system environment. Such an adjustment behavior was evidenced at the advent of the Bamako Initiative [59]. Other studies with varied designs reported that private providers offer more acceptable and convenient services [60, 61].

Secondly, predisposing factors, as stated by Anderson [21], including age, gender or filiation with household head, were not important factors influencing provider preference. The commonly speculated phenomenon of son-preference in provider choice was not found, although the phenomenon is a cultural legacy and openly voiced by scholars and the media in Burkina Faso. The findings may not be surprising as boys and girls receive immunization services indiscriminately, the most widely used health service. Furthermore, the urban context −where people are more educated and health literate−, fares well with better services and health advantages, therefore son-preference in healthcare may not be apparent. The phenomenon in rural areas may be different. There, girls still suffer from non-attendance, missed opportunities, and forced school- withdrawal in the education system. Ng'ambi et al. [62] have noted that youngest children (0–4 years old) were taken to formal providers that oldest (5–14 years old), as was urban children urban versus rural ones (AOR = 1.75, 95% CI = 1.12–2.72).

Thirdly, the study underscored the important role played by insurance coverage in formal care providers' utilization, this is consistent with prior research [63, 64]. The considerable proportion of insured children brought to private FP providers particularly argues that they offer better quality of care (competence, promptness...), as exhibited in Table 6. If extensive insurance did exist, the utilization of private care providers in Burkina Faso, would improve access to healthcare. As evidenced by the WHO [65, 66], universal insurance coverage plays a crucial role in improving access and equity in healthcare utilization by shielding households from a catastrophic financial burden.

Fourthly, factors such as socioeconomic status, education and occupation of household heads strongly influenced their formal provider choice. As expected, household heads with a higher education or a better job was associated with a greater likelihood of choosing a FP provider for their children. Better employment offers a great financial solvency to afford care, as FP providers are typically more expensive. Higher-educated household heads are generally more aware of where to seek better care for their children with regard to the condition encountered. In this study, instead of mothers' characteristics that we hypothesized to influence children HSB [67–69], the household head was favored, as culturally the household head (generally male) is the decision-maker which highlighted the patriarchal-oriented functioning of households in Africa.

Contrary to FP versus public providers, few significant associations were observed between FP versus public providers, as shown by Beogo et al. [70], on several aspects, there is a resemblance between public and NFP providers in Burkina Faso's healthcare delivery.

It is worth noting that, although it is widely recognized that the well-educated and well-paid jobholders prefer the private sector of care, very few studies in the literature have clearly delineated FP from NFP providers in their approach. As the private sector is very heterogeneous, FP and NFP providers differ significantly in their ownership, organization, management, financial incentives, and practices. Our findings highlight the importance in distinguishing among different types of private providers. A failure to do so otherwise, may compromise the usefulness of the findings, and misinform policy-making. In the same vein of the current results, in their previous study, in the same setting on adults HSB, Beogo et al. [70] noticed that as condition is perceived to be severe, intuitively, they clearly patronize formal providers seeking.

Our study has few limitations. First, several possible confounding factors, such as the disease episode outcome variables were not assessed. Future research which includes comprehensive and qualitative data may help to contribute in this regard. Second, this study was conducted in a typical urban area in SSA with an emerging private healthcare sector. Therefore, the findings may not be generalizable to suburban or rural areas. Third, data on income or wealth were not collected. Fourth, few studies were conducted in SSA that gathered data about insurance, which limited our ability to make feature comparisons. Finally, because of the correlated pattern of data, we implemented a GEE analysis, while previous studies utilized a multinomial logistic. Nevertheless, the GEE model appeared to be more robust, led to a comparatively very smaller standard error for all the imputed variables.

## Conclusion

This is one of few studies that looked at children healthcare seeking patterns in the context of an emerging private sector in an urban area in SSA. Although the health system still relies extensively on a public facilities' network, a significant portion of the urban pediatric population seeks private care providers in Burkina Faso. Capturing the presence of the private sector, particularly the FP providers, urges policy-makers to pay more attention to this rising group

associated with regulation issues [71]. Many LMICs countries, such as India, offer insightful examples. Private providers deliver both public health and curative services [60, 72]–up to 70% in urban areas [73]–allowing consumers to successfully navigate the coexisting systems to select services. Finally, as the private sector's provision of care thrives and is shown to reduce inequity in access to care, a strong partnership with its public counterpart would help to boost access to and utilization of the health system for children in SSA.

## Supporting information

**S1 File.**
(ZIP)

**S1 Table. Multilevel analysis using Poisson regression$^\varnothing$ for the choice between the formal providers by urban children by health condition.**
(DOCX)

## Acknowledgments

Thanks go to Dr Maxime Drabo for his support to shape the field strategy to collect the data. Our thanks go also to Caroline Kabiru for her contribution to the language edition.

## Author Contributions

**Conceptualization:** Idrissa Beogo.

**Data curation:** Idrissa Beogo, André Côté.

**Formal analysis:** Idrissa Beogo, Drissa Sia, Eric Tchouaket Nguemeleu.

**Funding acquisition:** André Côté.

**Methodology:** Idrissa Beogo, Drissa Sia, André Côté, Eric Tchouaket Nguemeleu.

**Software:** Idrissa Beogo.

**Supervision:** Drissa Sia, Patricia Bourrier, Darcelle Vigier, Nebila Jean-Claude Bationo, André Côté.

**Validation:** Patricia Bourrier, Darcelle Vigier, Eric Tchouaket Nguemeleu.

**Writing – original draft:** Idrissa Beogo, André Côté.

**Writing – review & editing:** Drissa Sia, Patricia Bourrier, Darcelle Vigier, Nebila Jean-Claude Bationo, André Côté, Eric Tchouaket Nguemeleu.

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
