## [Decision Letter · Decision Letter 0]

5 Jan 2022

PONE-D-20-08869Urban children health-seeking behavior in the context of free market: Household study in Ouagadougou, Burkina FasoPLOS ONE

Dear Dr. Beogo,

Thank you for submitting your manuscript to PLOS ONE. After careful consideration, we feel that it has merit but does not fully meet PLOS ONE’s publication criteria as it currently stands. Therefore, we invite you to submit a revised version of the manuscript that addresses the points raised during the review process.

 Two reviewers have evaluated your manuscript, and have identified a number of concerns that need to be carefully addressed in a revision. Please pay particular attention to the reviewers' recommendations regarding the contextualisation of your study and justification of the data treatment and statistical analyses.

We look forward to receiving your revised manuscript.

Kind regards,

Jamie Males

Staff Editor

PLOS ONE

Journal Requirements:

This paper is part of a doctoral project for which the author received a fellowship from Taiwan-ICDF and a research grant from the International Health Program of National Yang Ming University.

Reviewers' comments:

Reviewer's Responses to Questions

**Comments to the Author**

1. Is the manuscript technically sound, and do the data support the conclusions?

Reviewer #1: Yes

Reviewer #2: Partly

2. Has the statistical analysis been performed appropriately and rigorously? 

Reviewer #1: Yes

Reviewer #2: No

3. Have the authors made all data underlying the findings in their manuscript fully available?

Reviewer #1: Yes

Reviewer #2: Yes

4. Is the manuscript presented in an intelligible fashion and written in standard English?

Reviewer #1: Yes

Reviewer #2: Yes

5. Review Comments to the Author

Reviewer #1: Comments:

Methods:

The question for asking the health seeking behavior: is it based on a hypothetical scenario or based on a real case?

Have all children experienced all the conditions asked, the emergency, severe or non-severe? Is there any question to make sure that the children have experienced all that conditions? Do you have data about the diseases that the children have had? Please elaborate more.

The authors write that: for the purpose of this paper, self-medication and traditional healers were aggregated into the single category "informal provider". Can you elaborate more, perhaps in the introduction, how the position of traditional providers in the health care system in SSA?

Discussion:

Based on your data, can you discuss the differences between the pattern of caregiver’s HSB for children and the pattern of adults HSB? I see that you have published an article about the HSB for adults.

As in other country and cultural setting, the pattern for children HSB is different than that of adults as in this article:

Widayanti AW, Green JA, Heydon S, Norris P. Health-Seeking Behavior of People in Indonesia: A Narrative Review. J Epidemiol Glob Health. 2020 Mar;10(1):6-15. doi: 10.2991/jegh.k.200102.001.

Reviewer #2: Paper: Urban children health-seeking behavior in the context of free market: Household study in Ouagadougou, Burkina Faso

1. The title is very broad and year is missing. I would suggest a title “A cross-sectional study on health seeking behaviour amongst children from urban Ouagadougou in Burkina Faso: 2011” or “Factors associated with health seeking behaviour amongst children from urban Ouagadougou in Burkina Faso: 2011”

2. The data looks very old and outdated as it is almost 10 years older. The HSB may have changed in Ouagadougou in Burkina Faso.

3. In the abstract the authors do not come out very clearly about what conditions they are investigating. For example the following two papers (https://www.ncbi.nlm.nih.gov/pmc/articles/PMC8364791/ or https://onlinelibrary.wiley.com/doi/full/10.1111/tmi.13499 ) are clear on what was being investigated and what new knowledge they are contributing to HSB. I would recommend citation of https://onlinelibrary.wiley.com/doi/full/10.1111/tmi.13499 by the authors since this is amongst the few papers focusing on many diseases amongst the children.

4. The authors used GEE but a random effects model is better choice.

5. The introduction is too long. The authors should reduce it to two or three paragraphs

6. This study is more of secondary data analysis from the original study done in 2011, the methodology has been written as though the authors recently conducted the study design. The authors should reflect the secondary data analysis aspect of their study and revise the methods in that respect.

7. Health seeking behaviour is not clearly defined in the paper. The authors should consider reading this paper: https://onlinelibrary.wiley.com/doi/full/10.1111/tmi.13499

8. The data came from a complex survey but the data analysis section talks nothing on how complex design was taken into account in the analysis.

9. Why did the authors perform the analysis in three different software? Just one software suffices.

10. The analysis section does not specify the statistical model was used in the “GEE” yet the results present ORs

11. It is also not clear how the regression models were arrived at although there is a mention of sensitivity analysis

12. What was the level of statistical significance?

13. Table 2 talks about severity of illnesses. The missing piece is what was the illness? It is very misleading to just focus on illness severity.

6. PLOS authors have the option to publish the peer review history of their article (what does this mean?). If published, this will include your full peer review and any attached files.

Reviewer #1: No

Reviewer #2: **Yes: **Wingston Felix Ng'ambi

---

## [Author Response · Author response to Decision Letter 0]

9 Jan 2022

Response to Reviewers

Editorial office

Comment:

Response:

Done.

Comment:

This paper is part of a doctoral project for which the author received a fellowship from Taiwan-ICDF and a research grant from the International Health Program of National Yang Ming University.

Response:

Done according to your suggestion.

Comment:

Response:

Done according to your suggestion.

Comment:

Response:

The author received a fellowship from Taiwan-ICDF and a research grant from the International Health Program of National Yang Ming University. The publication fees were supported by the Faculty of Health Sciences, University of Ottawa. The funders had no role in study design, data collection and analysis, decision to publish, or preparation of the manuscript.

Comment:

Response:

This information is included in the cover letter 

Comment:

Response:

Done according to your suggestion.

Reviewer #1 

Comment (Methods):

The question for asking the health seeking behavior: is it based on a hypothetical scenario or based on a real case?

Response:

Thank you for this comment.

As you can see in the data collection tool, the questions #8, #11, and #14 are framed like: Where does [name of the household member] seek care in case of emergency matter? (See the questionnaire). The questions were not framed in hypothetical perspective. Instead, we sought their usual health-care-seeking behaviour, including their source of care in the 3 scenarios (Emergency, Severe, and not severe).

The question #33. “Could you tell me if you have suffered from any disease/injury in the past 30 days?” was reframed toward the end in order to validate their statement.

Indeed, this still does not rule out the potential bias, the recall bias for instance.

Comments:

Have all children experienced all the conditions asked, the emergency, severe or non-severe? 

Is there any question to make sure that the children have experienced all that conditions? Do you have data about the diseases that the children have had? Please elaborate more.

Response:

Thanks this important comment.

Yes, there was a refined question regarding an experience of disease. 

However, we noted that about 540 children have experienced a disease condition. I published on that question (Beogo, I., Huang, N., Gagnon, MP. et al. Out-of-pocket expenditure and its determinants in the context of private healthcare sector expansion in sub-Saharan Africa urban cities: evidence from household survey in Ouagadougou, Burkina Faso. BMC Res Notes 9, 34 (2016). https://doi.org/10.1186/s13104-016-1846-4). 

As the question was asking any disease experience of the past 30 days, obviously, we could not expect them to experience all the three conditions in this time frame (30 days). That was the main motive to look for their health-seeking behavior in case any of the three conditions occurred. 

Comment:

The authors write that: for the purpose of this paper, self-medication and traditional healers were aggregated into the single category "informal provider". Can you elaborate more, perhaps in the introduction, how the position of traditional providers in the health care system in SSA?

Response:

In the SSA health system, although traditional providers play an important role as a resort in health providing, they are not part of the formal health system, not plotted in the health pyramid. 

An excerpt was added in order to offer more light to the reader (p. 5). 

Comments:

Discussion:

Based on your data, can you discuss the differences between the pattern of caregiver’s HSB for children and the pattern of adults HSB? I see that you have published an article about the HSB for adults.

As in other country and cultural setting, the pattern for children HSB is different than that of adults as in this article:

Widayanti AW, Green JA, Heydon S, Norris P. Health-Seeking Behavior of People in Indonesia: A Narrative Review. J Epidemiol Glob Health. 2020 Mar;10(1):6-15. doi: 10.2991/jegh.k.200102.001.

Response:

As suggested, an excerpt was added to light more readers on the patterns of HSB for children versus adults. 

Please see on page 17.

Reviewer #2 

Comments:

1. The title is very broad, and year is missing. I would suggest a title “A cross-sectional study on health seeking behaviour amongst children from urban Ouagadougou in Burkina Faso: 2011” or “Factors associated with health seeking behaviour amongst children from urban Ouagadougou in Burkina Faso: 2011”

Response:

Thanks for your comment. As suggested, we include the period of the study (2011) in the study title.

We also took the opportunity of your comment to reshape the title in the light of the Burkina Faso context at the data collection period.

Comments:

2. The data looks very old and outdated as it is almost 10 years older. The HSB may have changed in Ouagadougou in Burkina Faso.

Response:

Yes, indeed, the number of privates facilities have augmented in Ouagadougou, since. However, no recent study has repeated the exercise. Therefore, we lack evidence to hypothesize, as the main indicators (e.g., poverty rate, 2014 in 2014 [See Kobiané, 2020]) have not change significantly.

Kobiané, J., Ouili, I. & Guissou, S. (2020). Etat des lieux des inégalités multidimensionnelles au Burkina Faso. Dans : Jean-François Kobiané éd., État des lieux des inégalités multi-dimensionnelles au Burkina Faso (pp. 1-89). Paris Cedex 12: Agence française de développement. https://doi.org/10.3917/afd.zanfi.2020.01.0001"

Comments:

3. In the abstract the authors do not come out very clearly about what conditions they are investigating. For example the following two papers (https://www.ncbi.nlm.nih.gov/pmc/articles/PMC8364791/ or https://onlinelibrary.wiley.com/doi/full/10.1111/tmi.13499 ) are clear on what was being investigated and what new knowledge they are contributing to HSB. I would recommend citation of https://onlinelibrary.wiley.com/doi/full/10.1111/tmi.13499 by the authors since this is amongst the few papers focusing on many diseases amongst the children. by the authors since this is amongst the few papers focusing on many diseases amongst the children.

Response:

Thanks for this fundamental inquiry.

 In fact, this study question was not targeting a specific disease. In contrary to Ng’ambi et al. (2020a) and Ng’ambi et al. (2020b) data were collected on the providers sought versus the condition severity based on their past experience (past 30 days). However, the question focusing on many diseases amongst the children was exploited and published for both children and adult.

 I found the papers quite interesting for ma manuscript and cited. 

Comments:

4. The authors used GEE but a random effects model is better choice.

Response:

Thanks for this fundamental comment. Indeed, and intuitively, the two-stage sampling approach led to a statistical approach that takes into account the hierarchical structure of the data. The clustering design was set with the aim of powering the representativeness of the sample. 

In order to determine a potential role played by the Administrative Sector (level1) and household (level2), we ran a multilevel analysis. In running the GEE, we considered the common choice of Unstructured Correlation to the three others. We paid a wise attention to the sensitive issue of the statistical plan throughout the study implementation. At the design stage we discussed and kept in touch with Dr Liu Chieh-Yu, a biostatician and coauthor of two of my papers; the last publication stemmed from the same macro-project on healthcare-seeking-behaviour. Furthermore, Dr Maxime K. Drabo, a domestic researcher was consulted and has actively contributed set the field method and validated the questionnaires before and after the pretest. After the completion of the data collection, we were so lucky to re-discuss and validate the statistical strategy with Prof Kung-Yee Liang ―Who was the principal of my University and beyond, has coined the GEE with Scott L. Zeger. The first paper published in PLOS ONE adhered to similar statistical mindset. https://journals.plos.org/plosone/article?id=10.1371/journal.pone.0097521

Comments:

5. The introduction is too long. The authors should reduce it to two or three paragraphs

Response:

Done accordingly. The revised introduction counts 259 words.

Comment:

6. This study is more of secondary data analysis from the original study done in 2011, the methodology has been written as though the authors recently conducted the study design. The authors should reflect the secondary data analysis aspect of their study and revise the methods in that respect.

Response:

This is a part of the PI Doctoral thesis. He collected primary data from 1600 households. That is way, in the methods, we kept the main then characteristics of the study setting. For instance, we inform about the research ethics grant and kept the then administrative sectors distribution. This may help the reader, namely of Burkina Faso, to be cautioned of the findings. 

Comment:

7. Health seeking behaviour is not clearly defined in the paper. The authors should consider reading this paper: https://onlinelibrary.wiley.com/doi/full/10.1111/tmi.13499

Response:

Thanks for the suggestion. We found it very interesting and even used it to give more light on the HSB definition in the background.

Comment:

8. The data came from a complex survey but the data analysis section talks nothing on how complex design was taken into account in the analysis. 

Response:

We thank you for this comment the Reviewer for this critical statistics-related comment.

As suggested, in order to determine a potential role played by the Administrative Sector (level1) and household (level2), we ran a multilevel analysis. Only the implementation of Poisson regression resulted in converging. We did consider that even though results were obtained for Emergency and non-severe conditions. For severe conditions, it failed to converge, possibly due to effects of the data complex structure or overparameterization (Bates, 2015). 

An excerpt was inserted to illustrate the complexity of data structure the statistical plan.

Please refer to “Statistical analysis” subsection (page 9).

Comment:

9. Why did the authors perform the analysis in three different software? Just one software suffices.

Response:

This was just for transparency matter as I started with SPSS and completed the analyses with Stata and SAS. In the present version we removed the Stata.

Comment:

10. The analysis section does not specify the statistical model was used in the “GEE” yet the results present ORs

Response:

Thanks for this comment. I have tried my best to address this question when answering to the comment 8.

Comment:

11. It is also not clear how the regression models were arrived at although there is a mention of sensitivity analysis

Response:

 I am not clear with this question. However, 

Comment:

12. What was the level of statistical significance?

Response:

I sorry that in this study I opted to not present the p-value. With my previous manuscripts, I used to meet reviewers who disagreed with p-value presentation. That is the only reason only the Confidence Interval is exhibited. 

Comment:

13. Table 2 talks about severity of illnesses. The missing piece is what was the illness? It is very misleading to just focus on illness severity.

Response:

Thank you for this comment.

The study did not focus on a specific disease. The questions were not framed in hypothetical perspective. We sought their usual health-care-seeking behaviour, including their source of care in the 3 scenarios (Emergency, Severe, and not severe).

In order to validate that quiring approach, there was a refined question regarding an experience of disease in the past 30 days. 

I published on that question (https://bmcresnotes.biomedcentral.com/articles/10.1186/s13104-016-1846-4).

---

## [Decision Letter · Decision Letter 1]

9 Jun 2022

PONE-D-20-08869R1Factors associated with health-seeking behavior amongst children in the context of free market: Household study in Ouagadougou, Burkina Faso, 2011PLOS ONE

Dear Dr. Idrissa Beogo,

Thank you for submitting your manuscript to PLOS ONE. After careful consideration, we feel that it has merit but does not fully meet PLOS ONE’s publication criteria as it currently stands. Therefore, we invite you to submit a revised version of the manuscript that addresses the points raised during the review process.

Address the outstanding comments below. Please submit your revised manuscript by 24 July, 2022. If you will need more time than this to complete your revisions, please reply to this message or contact the journal office at plosone@plos.org. Please include the following items when submitting your revised manuscript:A rebuttal letter that responds to each point raised by the academic editor and reviewer(s). You should upload this letter as a separate file labeled 'Response to Reviewers'.A marked-up copy of your manuscript that highlights changes made to the original version. You should upload this as a separate file labeled 'Revised Manuscript with Track Changes'.An unmarked version of your revised paper without tracked changes. You should upload this as a separate file labeled 'Manuscript'.If applicable, we recommend that you deposit your laboratory protocols in protocols.io to enhance the reproducibility of your results. Protocols.io assigns your protocol its own identifier (DOI) so that it can be cited independently in the future. For instructions see: https://journals.plos.org/plosone/s/submission-guidelines#loc-laboratory-protocols. Additionally, PLOS ONE offers an option for publishing peer-reviewed Lab Protocol articles, which describe protocols hosted on protocols.io. Read more information on sharing protocols at https://plos.org/protocols?utm_medium=editorial-email&utm_source=authorletters&utm_campaign=protocols.

We look forward to receiving your revised manuscript.

Kind regards,

Folusho Mubowale Balogun

Academic Editor

PLOS ONE

Journal Requirements:

Reviewers' comments:

Reviewer's Responses to Questions

**Comments to the Author**

1. If the authors have adequately addressed your comments raised in a previous round of review and you feel that this manuscript is now acceptable for publication, you may indicate that here to bypass the “Comments to the Author” section, enter your conflict of interest statement in the “Confidential to Editor” section, and submit your "Accept" recommendation.

Reviewer #2: All comments have been addressed

Reviewer #3: All comments have been addressed

2. Is the manuscript technically sound, and do the data support the conclusions?

Reviewer #2: Yes

Reviewer #3: Yes

3. Has the statistical analysis been performed appropriately and rigorously? 

Reviewer #2: Yes

Reviewer #3: Yes

4. Have the authors made all data underlying the findings in their manuscript fully available?

Reviewer #2: Yes

Reviewer #3: Yes

5. Is the manuscript presented in an intelligible fashion and written in standard English?

Reviewer #2: Yes

Reviewer #3: Yes

6. Review Comments to the Author

Reviewer #2: All comments have been addressed. Thank you authors for putting up a nice piece of work that is likely to improve the health system in Burkina Faso.

Reviewer #3: This manuscript is well written. However, the following minor issues need to be addressed:

1. In the abstract section, it is not clear what the reference category is University-educated household heads are compared in lines 16 to 19 on page 13.

2. On page 6 and line 9, the sample size formula is better described as “formula for estimating single proportion”, not “simple random sampling formula”.

3. On page 6 and line 21, The word “retained” should be replaced with “selected”.

4. On page 6 and line 21, the word “and” should be replaced with “or”.

5. On page 13 and line 20, delete the phrase “In conclusion”.

7. PLOS authors have the option to publish the peer review history of their article (what does this mean?). If published, this will include your full peer review and any attached files.

Reviewer #2: **Yes: **Wingston Felix Ng'ambi

Reviewer #3: No

---

## [Author Response · Author response to Decision Letter 1]

10 Jun 2022

Response to Reviewers

Reviewer #3

Comment:

1. In the abstract section, it is not clear what the reference category is University-educated household heads are compared in lines 16 to 19 on page 13.

Response:

Thanks for your comment. 

To improve the reading flow, we explicitly added the reference group, that is participants choosing public provider.

Comment: 

2. On page 6 and line 9, the sample size formula is better described as “formula for estimating single proportion”, not “simple random sampling formula”.

Response:

Thanks for your comment.

Correction done accordingly.

Comment: 

3. On page 6 and line 21, The word “retained” should be replaced with “selected”.

Response:

Correction done as suggested.

Comment: 

4. On page 6 and line 21, the word “and” should be replaced with “or”.

Response:

Thanks for this comment.

I could not figure out why ‘or’ fits better. Nevertheless, to improve the structure of the sentence and clarify the idea, two sentences are formed.

Comment: 

5. On page 13 and line 20, delete the phrase “In conclusion”.

Response:

Done as recommended

Thanks

Idrissa Beogo, PhD

---

## [Editor Report · Decision Letter 2]

4 Jul 2022

Factors associated with health-seeking behavior amongst children in the context of free market: Household study in Ouagadougou, Burkina Faso, 2011

PONE-D-20-08869R2

Dear Dr. Beogo,

We’re pleased to inform you that your manuscript has been judged scientifically suitable for publication and will be formally accepted for publication once it meets all outstanding technical requirements.

Kind regards,

Folusho Mubowale Balogun

Academic Editor

PLOS ONE
---

## [Editor Report · Acceptance letter]

23 Aug 2022

PONE-D-20-08869R2 

Factors associated with health-seeking behavior amongst children in the context of free market: Household study in Ouagadougou, Burkina Faso, 2011 

Dear Dr. Beogo:

I'm pleased to inform you that your manuscript has been deemed suitable for publication in PLOS ONE. Congratulations! Your manuscript is now with our production department. 

Kind regards, 

on behalf of

Dr. Folusho Mubowale Balogun 

Academic Editor

PLOS ONE